# Don't Trust ChatGPT when your Question is not in English: A Study of Multilingual Abilities and Types of LLMs

**Xiang Zhang**[*]  **Senyu Li**[*]  **Bradley Hauer**  **Ning Shi**  **Grzegorz Kondrak**
Alberta Machine Intelligence Institute
Department of Computing Science
University of Alberta, Edmonton, Canada
{xzhang23,senyu,bmhauer,ning.shi,gkondrak}@ualberta.ca

## Abstract

Large language models (LLMs) have demonstrated exceptional natural language understanding abilities, and have excelled in a variety of natural language processing (NLP) tasks. Despite the fact that most LLMs are trained predominantly on English, multiple studies have demonstrated their capabilities in a variety of languages. However, fundamental questions persist regarding how LLMs acquire their multilingual abilities and how performance varies across different languages. These inquiries are crucial for the study of LLMs since users and researchers often come from diverse language backgrounds, potentially influencing how they use LLMs and interpret their output. In this work, we propose a systematic way of qualitatively and quantitatively evaluating the multilingual capabilities of LLMs. We investigate the phenomenon of cross-language generalization in LLMs, wherein limited multilingual training data leads to advanced multilingual capabilities. To accomplish this, we employ a novel prompt back-translation method. The results demonstrate that LLMs, such as GPT, can effectively transfer learned knowledge across different languages, yielding relatively consistent results in translation-equivariant tasks, in which the correct output does not depend on the language of the input. However, LLMs struggle to provide accurate results in translation-variant tasks, which lack this property, requiring careful user judgment to evaluate the answers.

## 1 Introduction

The study of bilingualism has long been a topic of interest among linguists (Yu et al., 2022; Hoffmann, 2014), as it provides insight into the mechanisms of language acquisition and processing. Furthermore, research on multilingualism has contributed to the development of more effective machine learning models, such as neural translation systems (Zou et al., 2013). With the rise of large language models

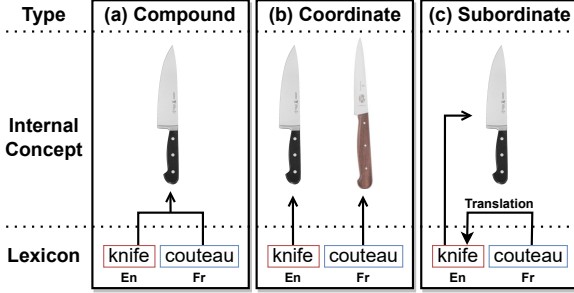

Figure 1: The three types of bilingualism.

(LLMs), researchers have discovered many emergent properties (Wei et al., 2022a) in these models, and have used them for a variety of purposes (Wei et al., 2022b). However, the multilingual ability of these models has not been extensively studied.

Previous research has shown that large language models, such as GPT, are capable of performing a wide variety of language tasks when the task is presented in English (Qin et al., 2023). However, investigations into the multilingual language abilities of these models have been limited. Shi et al. (2023) explore this topic by applying the models to multilingual datasets, and measuring performance differences across languages. However, they do not explore the underlying mechanisms of how LLMs perform different tasks, nor how this affects the results. Moreover, most LLMs (Brown et al., 2020; Touvron et al., 2023) are trained on datasets that are heavily skewed towards English, which leaves open the question of how multilingual abilities in such models are acquired.

In this study, we present a systematic approach to analyzing the multilingual capabilities of LLMs. To facilitate a comprehensive analysis, we propose categorizing language-dependent abilities into three distinct categories which vary in the impact of language choice on the performance: Reasoning (least impact), Knowledge Access, and Articulation (most impact). We investigate a carefully selected set of tasks from these three categories by evalu-

---

*Equal contribution.

ating the multilingual abilities of an LLM using a novel prompting method which we call response back-translation (RBT). By comparing the generated answers, we can both measure multilingual performance of the LLM, but also determine the type of multilinguality they exhibit. For example, we examine the capabilities of LLMs on pun detection, a highly language-dependent task.

The results of our experiments show that the popular LLM "GPT": (1) achieves higher performance when the task is presented in English; (2) achieves higher performance on tasks that can be translated without altering the correct output; and (3) exhibits a mixture of *coordinate* and *subordinate* bilingualism.

Our main contributions[1] are:

- We present a first-of-its-kind quantitative and qualitative analysis of the multilingual abilities of LLMs.

- We propose two novel task categorizations to facilitate the multilingual ability analysis.

- Our work is the first to investigate LLMs with respect to a linguistic typology of bilingualism and multilingualism.

## 2 Background

Linguists categorize bilingual individuals into three groups: compound, coordinate, and subordinate bilinguals (D'Acierno, 1990). Figure 1 illustrates this categorization, showing how individuals with different types of English-French bilingualism might internally represent the concept of "knife".

*Compound* bilingualism mostly emerges among individuals who learn two languages simultaneously from birth. In this case, both languages are equally dominant and integrated, blurring any clear distinction between them and giving the impression of a single unified language (Moradi, 2014). Compound bilingualism entails a shared mental representation of lexicons across both languages they acquire, and compound bilinguals are the most flexible in their use of multiple languages, exhibiting the ability to switch between languages without losing consistency in linguistic tasks (De Groot and Nas, 1991).

In contrast, individuals exhibiting *coordinate* bilingualism maintain separate mental representations for the lexicon of each language they

---

[1]Our data is publicly available at GitHub.

learn. This separation leads to differences when tasks are performed under different language settings (Jakobovits, 1968).

Finally, *subordinate* bilingualism is characterized by a "translator" behaviour (Marcos, 1976). This type of bilingualism is characterized by a single lexicon representation that is linked to their dominant language (Lörscher, 2012). When performing tasks in languages other than their dominant one, subordinate bilinguals tend to rely on translating the task into their dominant language, formulating an answer in the dominant language, and then translating that answer back into the language of the task. As a result, subordinate bilinguals may experience lower proficiency in communicating and completing tasks in the second, subordinate language.

Despite the demonstration in prior work of consistent multilingual performance in many large language models (Shi et al., 2023), it remains unclear how the multilingualism of LLMs should be categorized. It is an open question whether the LLMs exhibit a representation of knowledge shared across both languages (compound), separate representations for each language (coordinate), or whether they rely on a translation processes (subordinate). We develop an experimental framework aimed at using performance on various natural language processing tasks to determine how the multilingual abilities of LLMs relate to these categories.

## 3 Categorizing Language-Dependent Tasks

Language ability is a multifaceted concept encompassing various tasks and aspects (Wei et al., 2022a). It is therefore difficult to assess a model's capabilities with respect to a given language. To facilitate such assessment, researchers have often classified tasks into distinct categories (Khurana et al., 2023), such as parsing and summarization. However, the delineation of such categories often lacks systematic criteria, particularly in the context of multilingual analysis.

In this section, we propose a novel approach to categorizing NLP tasks, which is better suited to analysis of multilingual abilities. The categorization is two dimensional: one dimension is based on the linguistic knowledge necessary to complete the task (Section 3.1), the other on how the task is impacted by the language in which it is presented (Section 3.2).

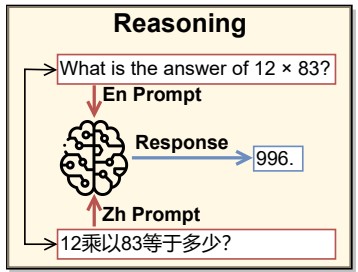
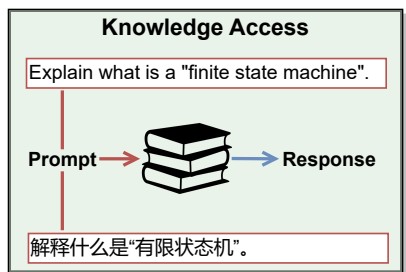
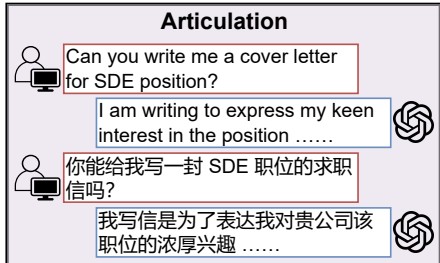

Figure 2: Three categories of NLP tasks.

## 3.1 Categorization by Task Properties

We classify NLP tasks into three distinct categories: Reasoning, Knowledge Access, and Articulation. This division is based on the extent to which performance on each task is influenced by the model's capabilities with the language used. Figure 2 provides an overview of this categorization.

**Reasoning** The first category includes tasks that are minimally influenced by language, on which consistent performance is expected across languages. Reasoning tasks involve logical and rational thinking to solve problems based on available information and logical principles. Examples include mathematical problem-solving (Lu et al., 2023), coding (Li et al., 2023), and common sense reasoning (Sap et al., 2020). These tasks can be performed using universal language elements, such as mathematical symbols, or rely on general life experience and common sense, which can be acquired without language. For example, answering the question *"If I drop an apple, which direction will it go?"* relies more on understanding gravity than on language-specific knowledge.

**Knowledge Access** LLMs have the capability to function as knowledge bases (KBs) by storing knowledge extracted from training data (Heinzerling and Inui, 2021). Knowledge Access tasks depend on the ability to access this knowledge and formulate accurate responses based on it. While the underlying knowledge may not be language dependent, models may be less reliable in retrieving and utilizing knowledge learned in a language other than the one used to formulate the task. Examples of Knowledge Access tasks include factual knowledge checking (De Cao et al., 2021), knowledge-focused question answering (Wang, 2022), and named entity recognition (Malmasi et al., 2022).

**Articulation** Much of everyday human conversation is highly language-dependent, as it involves the pragmatics and cultural nuances of the spoken language. For instance, writing a cover letter in English significantly differs from writing one in Japanese, due to the distinct social norms and conventions associated with those languages. The Articulation category includes tasks that are heavily influenced by the language choice, such as summarization (Nenkova and McKeown, 2012), dialogue generation (Ni et al., 2023), paraphrasing (Zhou and Bhat, 2021), and style writing (Jin et al., 2022). These tasks require an extensive understanding of not only language, but the associated culture, as they involve capturing and reproducing the appropriate style, tone, and manner of expression specific to a given language.

## 3.2 Categorization by Translatability

The second dimension of our task classification scheme involves translatability. We introduce the concepts of Translation Equivariant (TE) and Translation Variant (TV) tasks.

A function is considered *equivariant* if it commutes with a symmetry transformation. That is, applying a transformation before or after computing the function yields the same result. Formally , $f(\cdot)$ is said to be equivariant under $g(\cdot)$ if:

$$\forall x \in \mathcal{D}, \quad g(f(x)) = f(g(x)) \tag{1}$$

where $\mathcal{D}$ represents the domain of both $f$ and $g$.

We denote translation as a transformation $g$ that converts a given text in language A to an equivalent text in language B. In practice, $g$ can be implemented by a machine translation system. We further use $f$ to denote a function which solves a given task, given an instance of that task as input. A task is considered Translation Equivariant between languages A and B if the correct output can be obtained by translating the input, and then applying a method for solving the task, or by solving the task, and then translating the output; in other words, if $g(f(x)) = f(g(x))$. Most of the

tasks in the Reasoning and Knowledge Access categories are regarded as Translation Equivariant since the correct output does not depend on the chosen language. Figure 3 shows an example where the answer to the question posed in English remains the same in Chinese, regardless of in which order the translation system and the question answering system are applied.

A task which is not Translation Equivariant is Translation Variant. For such tasks, translating the input may change the correct output. TV tasks rely heavily on the language used, and include many tasks in the Articulation category. Representative TV tasks that we investigate in our experiments are *letter writing* and *pun understanding*. The former is subject to the conventions of the specific language and culture, while the latter involves word polysemy, which is often sensitive to translation. Figure 3 shows an example where a pun is present in the original English input, but not in the Spanish translation, making the classification dependent upon the order in which translation is applied.

## 4 Methods

In this section, we present our approach to analyzing the multilingual ability of LLMs. Our methods involve *prompt translation* (PT) and *response back-translation* (RBT). They are designed to measure performance of an LLM, and its consistency across languages. In our experiments, we apply these methods to both TE and TV tasks, with the aim of determining the type of bilingualism (compound, coordinate, or subordinate) exhibited by an LLM.

### 4.1 Prompt Translation

Multilingual datasets are unvailable for many tasks. However, with state-of-the-art machine translation (MT) systems and LLMs, we can translate monolingual datasets for TE tasks to generate parallel multilingual parallel data with minimal loss of information (Whitehouse et al., 2023; Shi et al., 2023). This is the key intuition behind prompt translation (PT); an example is shown in Figure 4a, where an English multiple choice question, and its possible answers, are translated to Chinese. The LLM is then prompted, and the response is given and evaluated, in Chinese. Prompting in distinct languages is performed in independent LLM sessions.

We measure the differences in multilingual task performance by comparing the answers given by the LLM in each language. Assuming that the LLM

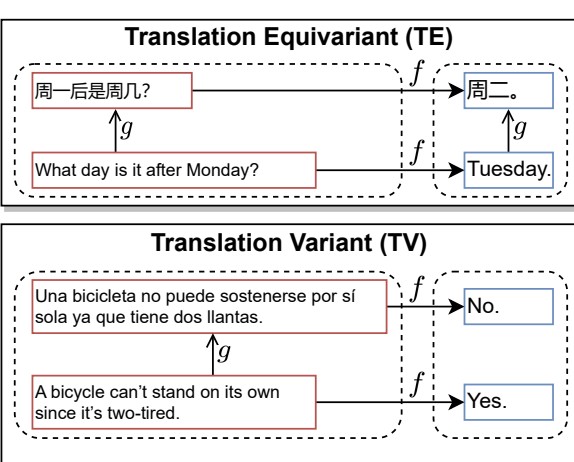

Figure 3: A TE task (common sense reasoning) and a TV task (pun detection). Translation is denoted by $g$, and $f$ is the solver function.

successfully learns to solve a TE task in a language-independent way, the pairwise responses for each instance should be the same after the translation (regardless of whether it is correct or incorrect). This is because TE tasks, such as mathematical problem solving, do not depend on the language used to query the LLMs, as the solution does not depend on the language used to express the problem.

### 4.2 Response Back-Translation

One of the goals of our work is to understand what the consistency of LLM output across languages tells us about the model, and to determine the type of bilingualism an LLM exhibits. This is crucial for individuals who use LLMs for multilingual tasks, as it can impact the way task results are generated, and affect the quality and consistency of the results. For example, a network exhibiting subordinate bilingualism would produce output that appears to be the result of translation, rather than resembling text generated by a native speaker of the output language.

To quantitatively measure how reasoning is performed, we propose a prompting method based on back-translation, as illustrated in Figure 4b. Similar to prompt translation (Section 4.1), we begin by translating the instance to the target language, and prompting the LLM to produce a response in that language. After obtaining output from the LLM, regardless of the language, we further prompt the LLM to generate an explanation for its output (e.g., *"Explain how you obtain this result"*); and then translate the output of the LLM back to the original language. We then compare the explanation given

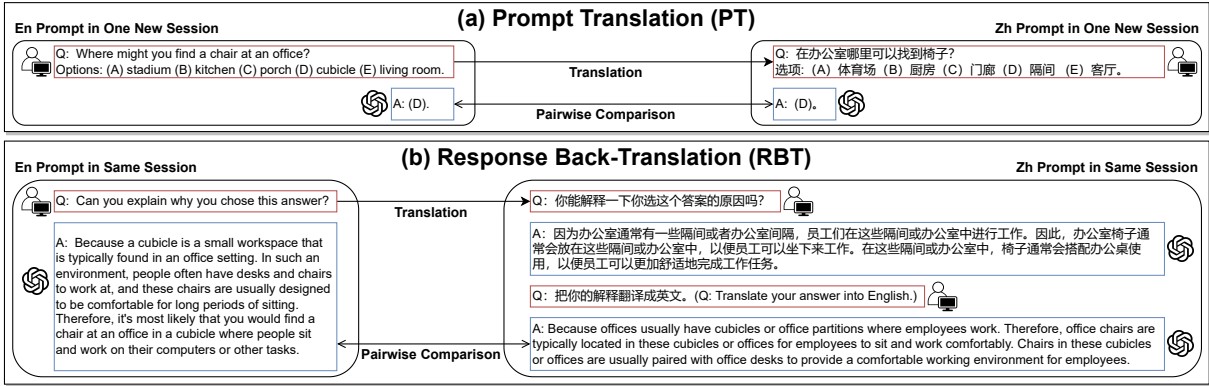

Figure 4: An overview of our prompt translation and response back-translation methods.

in the source language to the explanation back-translated from the target language.

If the LLM is performing translation-based reasoning, the reasoning process is conducted in one language and then translated into another. Since the internal reasoning of the LLM can be partially observed through the output explanation, back-translating such explanations into the source language allows us to compare the internal reasoning used to solve the problem in each language. High similarity of explanations should indicate homogeneity in using the same internal reasoning process to perform the task in both languages. On the other hand, dissimilarity in the reasoning process across languages should be reflected in a lower explanation similarity.

### 4.3 Identifying Multilingual Types

In our investigation, we employ both Prompt Translation (PT) and Response Back-Translation (RBT) to analyze how an LLM solves TE and TV tasks in different languages. As depicted in the first two steps in Figure 5, a compound LLM should exhibit consistent results on TE tasks with both methods. This is because a compound model performance does not depend on the language in which a question is presented. Conversely, subordinate and coordinate types of networks are expected to yield somewhat different results on TE tasks. A coordinate model accesses distinct representations in different languages, which may result in different reasoning and answers. Finally, a subordinate model heavily depends on an internal translation process, which we expect to lead to some deterioration of output quality across languages.

Testing on TV tasks provide additional information, which can be used to distinguish between coordinate and subordinate models. A coordinate LLM

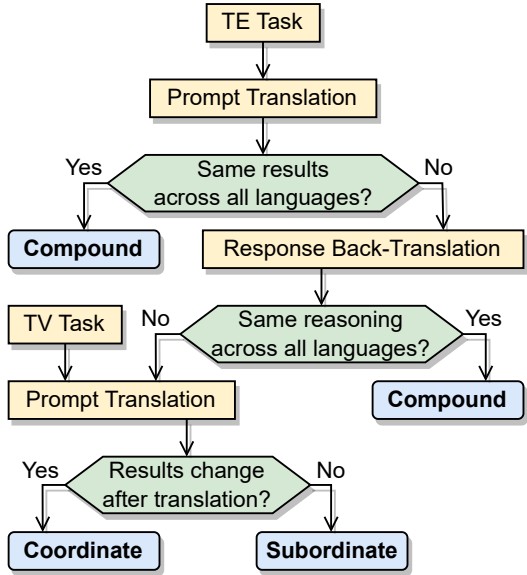

Figure 5: Flowchart for detecting multilingual types.

is expected to reason differently for each language, which may yield different outputs, whether correct or not. In contrast, a pure subordinate model is expected to reason only in the dominant language, producing relatively similar results in different languages, regardless of whether the correct output is preserved after translation.

## 5 Experiments

We apply the methodology proposed in Section 4 to TE and TV tasks. As our LLM, we use Chat-GPT, via the official web application[2], due to its availability.

### 5.1 Datasets

**Reasoning** We use 50 instances selected at random from each of two datasets: GSM8K (Cobbe

---

[2] https://chat.openai.com/

et al., 2021), which contains 7,500 training and 1,000 test problems, and CommonsenseQA (Talmor et al., 2019), which contains 12,247 questions. We used ChatGPT to translate these instances into French, Spanish, German, Japanese, and Chinese. GSM8K is a dataset of grade-school math problems. Each problem consists of a question and a multiple-choice answer. CommonsenseQA is a question answering dataset for testing logic and common sense. Each instance consists of a question and five answer choices, only one of which is considered correct.

**Knowledge Access** WebQuestions is a dataset of 6,642 question-answer pairs extracted from Freebase (Bordes et al., 2014). An example question is *"Where is the Thames River located?"* to which the correct answer is *London*. To simplify the evaluation, and avoid the issue of extracting answers from ChatGPT's often verbose responses, we manually converted 50 randomly selected instances into the multiple-choice format used by CommonsenseQA. To create plausible incorrect answers (*distractors*), we randomly selected four incorrect candidate answers from sets of world city names[3] and celebrity names[4] (correct answers in this dataset are all either city names or celebrity names). This yielded a set of 50 multiple choice questions with five possible answers each (one correct, four incorrect). We translated the English instances into five other languages via ChatGPT.

**Puns** We randomly selected 80 positive and 80 negative instances each from the English, French, and Spanish instances in the JOKER@CLEF 2022 dataset (Ermakova et al., 2022). Each instance is annotated with a yes/no classification as to whether it contains a pun, and the pun location, if a pun is present. An example English instance is *"Astronauts work in a nice atmosphere"* for which the pun location is the word *atmosphere*. We used ChatGPT to translate the French and Spanish instances into English, and the English instances into French, Spanish, German, Japanese, and Chinese. This yields 10 balanced sets of 160 instances each (three original and seven translated).

**Articulation** To test the Articulation abilities of an LLM, we prompt the model to generate a cover letter for a job application, with randomized specifications. For each prompt, we first generate the

[3] https://simplemaps.com/data/world-cities
[4] https://github.com/janester/mad_libs

| Task | **En** | Fr | De | Es | Ja | Zh |
|------|--------|------|------|------|------|------|
| MR | **0.90** | 0.80 | 0.78 | 0.80 | 0.82 | 0.78 |
| CSR | **0.68** | 0.58 | 0.52 | 0.54 | 0.48 | 0.52 |
| KA | **0.96** | **0.96** | 0.94 | 0.94 | 0.80 | 0.68 |

Table 1: Accuracy for TE tasks: math reasoning (MR), commonsense reasoning (CSR), and knowledge access (KA).

name and background of an individual, including information such as level of education, specialties, and hobbies. We then randomly select one well-known company to which cover letter is to be addressed. Finally, we select a set of topics such as "What skills would you want to develop in this role?". Each of these randomized prompts is then provided to the LLM. The output is then manually evaluated by a native speaker of the language of the prompt. We generate 50 prompts each in English and Chinese. An example is provided in Table 5 in the appendix.

## 5.2 Metrics

Since ChatGPT can give different answers to the same question, we present each multiple-choice question to ChatGPT five times, and use the most frequent output for evaluation. For computing similarity between explanations, we use appendix(Devlin et al., 2019). Specifically, we translate all non-English output to English via ChatGPT, and compute the cosine similarity of the BERT embeddings of the two explanations.

## 5.3 Results on TE Tasks

As shown in Table 1, the results on TE tasks in English are on average much higher in English than in other languages. In math reasoning (MR), the least language-dependent task, the gap between English and other languages is over 10% on average. In common sense reasoning (CSR), the difference is over 15% on average. In knowledge access (KA), there is no substantial difference between English and other European languages, but accuracy on Japanese and Chinese is 16% and 28% lower, respectively. To confirm that the accuracy gap is not due to instance translation quality, we manually compared all 50 Chinese MR questions with their original English counterparts, and found no translation errors. Taken together, these results provide strong evidence that GPT is better able to reason and retrieve knowledge given an English

prompt, compared to prompting in other languages. In terms of multilinguality type, the evidence is against compound multilingualism in GPT (cf., Figure 5), as a compound model would be expected to exhibit no substantial difference in performance across languages.

We also analyzed the BERT similarity values between explanations in different languages (cf., Table 4 in the Appendix). In commonsense reasoning, which relies on logic and conceptual distinctions, we observe that the average BERT similarity of German, Spanish, Japanese and Chinese to French is substantially lower than the corresponding average similarity to English (0.849 vs. 0.868), while French itself is substantially more similar to English than to German (0.871 vs 0.857). We interpret this as additional evidence of the GPT's dependence on its strong English model.

On the other hand, we observe no such trend in knowledge access questions. We hypothesize that since these problems are mostly about named entities, they tend to be more language-independent. Indeed, we observe higher performance on French, German, and Spanish, which use the Latin script, and therefore can represent named entities as English does, compared to Japanese and Chinese, which use different orthographies.

### 5.4 Cover Letters

Cover letter writing is an example of a TV articulation task. We found that cover letters generated by ChatGPT with the same set of instructions in different languages exhibit relatively high BERT similarity to their English versions, ranging from 0.818 Japanese to 0.865 for German. To provide some comparison, we also computed pairwise BERT similarities between English cover letters generated with the same prompts by ChatGPT and two other LLMs, Claude and Instant-Claude, which yielded the values of 0.618 and 0.643, respectively. This indicates that the letters generated in different languages by ChatGPT are more similar to each other than the letters in English generated by different LLMs.

Cover letters generated in languages other than English exhibit a written style which is closer to English than to the target language. For example, consider the cover letter shown in Table 5 in the Appendix. The expressions 阁下 "from what I have gathered" and 狂热的户外运动爱好者 "avid outdoor enthusiast" are very unnatural in Chinese,

| Chinese | English Translation | Frequency |
|---|---|---|
| 诚挚地 | Sincerely | 54.0% |
| 致意 | Regards | 38.4% |
| 祝愿 | Best Wishes | 3.6% |
| 此致敬礼 | Salute (Proper Chinese Sign-off) | 0.8% |
| No sign-off | | 3.2% |

Table 2: The frequency of different sign-offs in 250 different Chinese cover letters generated by ChatGPT.

| Language | P-Acc | L-Acc |
|---|---|---|
| Es | 0.488 | 0.697 |
| Es-En | 0.507 | 0.714 |
| Fr | 0.500 | 0.886 |
| Fr-En | 0.513 | 0.813 |
| En | 0.506 | 0.965 |
| En-Fr | 0.500 | 0.646 |
| En-De | 0.519 | - |
| En-Es | 0.488 | 0.607 |
| En-Ja | 0.519 | - |
| En-Zh | 0.550 | 0.511 |

Table 3: Accuracy on pun detection (P-Acc) and location (L-Acc). X-Y means the puns were translated from language X to language Y before prompting.

and appear characteristic of literal translations from English. The sign-off phrase 真诚的 "Sincerely" is similarly inappropriate in formal Chinese, although it is usual in English. Table 2 shows that less than 1% of the letters have a proper Chinese sign off.

### 5.5 Results on Puns

Table 3 shows the results on the translation-variant tasks of pun detection and location. The accuracy of pun detection is close to what we would expect from a random baseline, as ChatGPT strongly favors positive pun classifications. The sole exception is a slightly higher accuracy of 0.55 when English puns are translated into Chinese, due to a higher proportion of negative classifications.

Since few conclusions can be drawn from the pun detection results, we conducted an evaluation of the pun location results in most datasets, which required manual extraction of the location information from ChatGPT's explanations. The results are shown in Table 3. The pun location accuracy on the original English puns is very high at 96.5%, but drops dramatically when the sentences are translated into other languages. When French puns are translated to English, there is likewise a

drop in performance, though it is much smaller than what is observed when English puns are translated to French. However, the situation is different for Spanish puns, where the location accuracy increases slightly after the puns are translated into English. This is surprising, as puns are often language-specific, and tend to disappear after translation.

When the prompt is not in English, evidence suggests that ChatGPT relies, at least partly, on its English capabilities for semantic interpretation. Consider the homonymous English word *bat* which has two unrelated senses, translated by different words in Chinese: 蝙蝠 for the "animal" sense, and 球拍 for the "club" sense (Hauer and Kondrak, 2020). When the original English prompt is *"What is the famous bat brand for baseball?"*, ChatGPT appears unable to distinguish between these two translations of *bat* within a Chinese prompt. Although the choice of the Chinese translation of *bat* greatly affects the meaning of the question, it does not seem to impact ChatGPT's response. However, when we replace 蝙蝠 "animal bat" with 老虎 "tiger", ChatGPT correctly responds that the question makes no sense. We interpret the inability of ChatGPT to differentiate between the two distinct Chinese translations of *bat* as strong evidence of subordinate bilingualism.

### 5.6 Analysis of Results

The results of our experiments provide evidence that GPT exhibits a substantial degree of subordinate multilingualism. Many of its responses are what we would expect from a system which translates all input into English, formulates a response in English, and then translates this response into the input language. Since translation is an error-prone process, the resulting response accuracy is frequently lower than when the input is provided in English.

We speculate that this behavior is an artifact of GPT being trained mostly on monolingual English texts. Consequently, GPT has developed a representation of knowledge and communication that is strongly biased towards English. We conclude that since GPT is not designed to take advantage of bilingual or multilingual corpora, it is unable to create a single multilingual conceptual representation analogous to compound multilingualism.

Moreover, GPT has less training data for non-English languages, compared to its English training data. We postulate that this results in represen-

tations for non-English languages that are much weaker than those GPT can create for English. This often leads to lower performance on even translation-equivariant tasks when the task is not presented in English.

## 6 Discussion and Future Directions

Our research provides robust support for the notion that LLMs have not achieved the ideal behaviour of compound multilingualism. Even if the quality and quantity of training data in various languages were held constant, we speculate that compound multilingualism would still not be achieved, due to the inherent limitations of current data collection methods and training techniques.

Drawing a parallel to human multi-modal learning offers an intuitive understanding of why this could be the case. Consider how humans acquire concepts related to vision and language: A child grows by consistently pairing visual stimuli with linguistic cues, intertwining the two modalities over time. Consequently, it is rare to observe a mismatch between visual and linguistic perceptions. In this context, humans exhibit a highly integrated understanding of vision and their native languages. However, unless raised in a perfectly bilingual environment, individuals seldom showcase equivalent proficiency in two languages. Indeed, bilingual individuals often demonstrate cognitive variations depending on which language is in use.

A rudimentary multi-modal system can be likened to a crude fusion of a vision model trained on image data and a language model trained on text. These systems possess minimal, if any, shared representations or information overlap. Beneath the facade of a system that seemingly excels at both visual and language tasks, lie two distinct networks. Nevertheless, recent advancements in multi-modal studies, combined with the availability of extensively captioned image data, have given rise to more sophisticated systems. These systems bridge the gap between the two modalities, moving the field closer to human-like integration.

Acquiring aligned multilingual data is a significant challenge, with the exception of some translation datasets. The majority of online articles and posts are monolingual and cannot be easily paired. Therefore, training on these multilingual corpora results in models that essentially act as an amalgamation of several independent language-specific models, with minimal information interchange, pri-

marily anchored by the translation datasets which comprise a relatively small portion of the corpus. When corpora are disproportionately comprised of some language or set of languages, the models tend to become predominantly subordinate, with minimal coordination arising from monolingual datasets.

Moving forward, our objective is to narrow the divide between languages within a multilingual system and to cultivate language models that lean more towards a compound archetype. This will require both crafting highly parallel paired data across languages and innovating training methodologies that promote the learning of compound representations for universal concepts irrespective of the language used to express them. For the former, we intend to delve into ontology linkages. For the latter, we plan to leverage recent advancements in model training, such as contrastive learning. Our goal is to create multilingual models that are both technically sophisticated and universally adept.

## 7 Conclusion

We have proposed a systematic approach to analyzing multilingual abilities of large language models. Our experiments provide new evidence for a subordinate multilingualism in GPT-3.5, with English functioning as the model's native language. Our experimental results, supplemented by the analysis of specific examples and case studies, demonstrate that such subordinate multilingualism can limit performance even in language-independent tasks. We postulate that explicit inclusion of additional multilingual parallel corpora and multimodal datasets into the training data of LLMs could ameliorate this issue.

## Limitations

As the OpenAI ChatGPT website application has a limited number of prompts allowed per day and per hour, we can not apply our experiment to the whole dataset. We used GPT3.5 rather than GPT4 as our LLM since access to GPT4 is still restricted. We conducted the human evaluation only in English, Spanish, and Chinese, as we did not have access to fluent speakers of the other languages found in our test sets.

In addition to ChatGPT, we also attempted to apply a different LLM, Llama2 to the same tasks (Touvron et al., 2023). The outcomes from Llama2, however, were unexpected for several reasons.

Firstly, Llama2 frequently produced responses that lacked meaningful content, making answer extraction challenging. Secondly, Llama2 often declined to provide answers to posed questions (cf. Table 7 in the Appendix). Thirdly, when posed with questions in languages other than English, Llama2 usually responded in English. Lastly, inherent issues in the datasets, such as inconsistent capitalization and grammatical errors, further complicated the evaluation.

## Acknowledgements

This research was supported by the Natural Sciences and Engineering Research Council of Canada (NSERC), and the Alberta Machine Intelligence Institute (Amii).

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

# A   Appendix

The appendix contains four tables.

Table 4 contains BERT similarity scores between the explanations generated by ChatGPT in different languages (Section 5.3).

Table 5 contains a comparison of cover letters generated by ChatGPT in English and Chinese, respectively (Section 5.4).

Table 6 contains examples of the explanations of an English pun generated by ChatGPT in English, Chinese, French, and Spanish (Section 5.5). The pun in question is *"a bicycle can't stand on its own because it is two-tired"*, with a word-play involving the homophonic phrases *too tired* and *two-tire'd*. The pun is lost in translation to other languages, but this does not prevent CHatPGT from correctly detecting, locating, and explaining it in those languages.

Table 7 shows an example of a misguided response from Llama2.

| Lang. | En | Fr | De | Es | Ja |
|---|---|---|---|---|---|
| Common Sense Reasoning | | | | | |
| Fr | 0.871 | 1.000 | | | |
| De | 0.882 | 0.857 | 1.000 | | |
| Es | 0.886 | 0.864 | 0.868 | 1.000 | |
| Ja | 0.839 | 0.822 | 0.850 | 0.828 | 1.000 |
| Zh | 0.863 | 0.852 | 0.872 | 0.866 | 0.848 |
| Math Reasoning | | | | | |
| Fr | 0.947 | 1.000 | | | |
| De | 0.937 | 0.933 | 1.000 | | |
| Es | 0.948 | 0.947 | 0.940 | 1.000 | |
| Ja | 0.907 | 0.905 | 0.910 | 0.901 | 1.000 |
| Zh | 0.905 | 0.912 | 0.902 | 0.912 | 0.902 |
| Knowledge Access | | | | | |
| Fr | 0.928 | 1.000 | | | |
| De | 0.916 | 0.926 | 1.000 | | |
| Es | 0.925 | 0.922 | 0.903 | 1.000 | |
| Ja | 0.883 | 0.888 | 0.878 | 0.883 | 1.000 |
| Zh | 0.858 | 0.868 | 0.864 | 0.848 | 0.864 |

Table 4: BERT similarity scores for explanations in TE tasks.

| | |
|---|---|
| **Prompt(EN):** | You are Johnson Smith from University of Alberta with A GPA of 3.9. You like sapping. You want to join Huawei company. Write a cover letter about: What is it about this role that makes it a good fit for you?, What's something outside of your work that you're passionate about?, and What does your next ideal role look like? |
| **English:** | Dear Hiring Manager, |
| | I am writing to express my keen interest in joining Huawei Technologies Co. Ltd. as a [insert desired position]. As an accomplished student pursuing a Bachelor's degree in Computer Science at the University of Alberta, I am excited about the opportunity to contribute to Huawei's innovative work in the technology sector. |
| | What sets Huawei apart for me is its commitment to pushing the boundaries of technological advancement and delivering high-quality products and services to its global customers. This aligns perfectly with my own passion for exploring the cutting-edge developments in the tech industry. Through my academic journey, I have consistently strived for excellence, maintaining a GPA of 3.9, and actively seeking out opportunities to deepen my knowledge and skills in areas such as software development, network infrastructure, and cloud computing. |
| | I firmly believe that my technical expertise and problem-solving abilities make me an excellent fit for the role at Huawei. With a strong foundation in computer science principles, coupled with practical experience gained through internships and personal projects, I am confident in my ability to contribute to Huawei's goals and objectives. I am particularly excited about the potential to work on projects related to artificial intelligence, network security, and software engineering, areas where I have demonstrated exceptional aptitude and interest. |
| | Beyond my professional pursuits, I am a firm believer in maintaining a well-rounded lifestyle. One of my greatest passions outside of work is sapping. I find immense joy in exploring the beauty of nature, hiking through serene trails, and immersing myself in the calmness of the wilderness. Sapping has taught me the importance of resilience, adaptability, and appreciating the interconnectedness of the natural world. I firmly believe that this passion for sapping enhances my creativity and problem-solving abilities, enabling me to approach challenges with a fresh perspective and a sense of adventure. |
| | Looking forward, my ideal role involves being part of a dynamic and collaborative team that encourages innovation and continuous learning. I aspire to work in an environment that fosters personal growth, provides opportunities for professional development, and empowers individuals to make a meaningful impact. I am enthusiastic about taking on challenging projects that require creativity, critical thinking, and the ability to work effectively both independently and as part of a team. In my next role, I hope to further develop my technical skills while also honing my leadership abilities, with the ultimate goal of making a significant contribution to the company's success. |
| | Thank you for considering my application. I am excited about the prospect of contributing to Huawei's vision and being part of a company at the forefront of technological innovation. I have attached my resume for your review, and I look forward to the opportunity to discuss my qualifications further. |
| | Yours sincerely, |
| | Johnson Smith |
| **Prompt(Zh):** | 你是来自阿尔伯塔大学的约翰逊·史密斯，GPA为3.9。你喜欢程序开发。你希望加入华为公司。请撰写一封求职信，内容包括以下问题：这个职位有什么特点使其适合你？你工作之外的兴趣爱好是什么？你心目中的下一个理想职位是怎样的？ (in a completely new session) |
| **Chinese:** | 阁下， |
| | 我谨向华为公司递交我的求职信，表达我加入贵公司的热忱和兴趣。我是约翰逊·史密斯，毕业于匿名大学，拥有3.9的GPA。我对程序开发充满热情，并且相信华为公司是我继续发展和实现个人职业目标的理想之地。 |
| | 在我了解的情况下，华为公司在全球范围内的领先地位和不断创新的文化吸引了我。我对华为的技术解决方案和产品充满敬佩，并希望能够为公司的成功作出贡献。我在匿名大学的学习和项目经验为我提供了坚实的编程基础和解决问题的能力，这些技能将使我能够在日常工作中充分利用华为的技术平台和工具。 |
| | 除了工作之外，我对多种兴趣爱好保持着活跃的参与。我是一个狂热的户外运动爱好者，喜欢徒步旅行、露营和攀岩。这些活动锻炼了我的团队合作和领导能力，并培养了我在压力下保持冷静和解决问题的能力。此外，我也热衷于参与志愿者活动，尤其是与青少年教育和技术培训相关的项目。这些经历不仅增强了我的沟通和人际关系技巧，还培养了我对社会责任的意识。 |
| | 在我心目中，下一个理想职位是成为一名技术领导者。我希望在技术领域中不断成长和学习，领导并激励团队实现共同目标。我相信通过不断挑战自己，我能够发挥我的领导潜力并在组织中产生积极的影响。 |
| | 我期待着有机会加入华为公司，并为实现公司的愿景和目标而努力工作。感谢您抽出宝贵的时间考虑我的申请。我期待着与您进一步讨论我对这个职位的适合性。 |
| | 谢谢您的关注。 |
| | 真诚地, |
| | 约翰逊·史密斯 |
| **Prompt(En):** | Translate this into English. |
| **English:** | Dear Sir/Madam, |
| | I am writing to express my keen interest in joining Huawei Company and hereby submit my application. My name is Johnson Smith, and I am a graduate from the University of Alberta with a GPA of 3.9. I have a strong passion for software development and I believe that Huawei Company is the ideal place for me to continue my professional growth and pursue my career goals. |
| | From what I have gathered, Huawei Company's leading position worldwide and its culture of continuous innovation have captivated me. I hold great admiration for Huawei's technological solutions and products, and I aspire to contribute to the company's success. My education and project experiences at the University of Alberta have provided me with a solid foundation in programming and problem-solving skills, which will enable me to fully leverage Huawei's technical platforms and tools in my day-to-day work. |
| | Outside of work, I actively engage in various hobbies and interests. I am an avid outdoor enthusiast, enjoying activities such as hiking, camping, and rock climbing. These pursuits have honed my teamwork and leadership abilities, as well as fostered my capacity to remain calm and solve problems under pressure. Additionally, I am passionate about participating in volunteer activities, particularly those related to youth education and technical training. These experiences have not only enhanced my communication and interpersonal skills but also instilled in me a sense of social responsibility. |
| | In my vision, my next ideal position is to become a technical leader. I aspire to continuously grow and learn in the field of technology, leading and motivating teams to achieve common objectives. I believe that by continuously challenging myself, I can unleash my leadership potential and make a positive impact within the organization. |
| | I am eagerly looking forward to the opportunity to join Huawei Company and contribute to the realization of its vision and goals. Thank you for taking the time to consider my application. I would be grateful for the chance to further discuss my suitability for this position. |
| | Thank you for your attention. |
| | Sincerely, |
| | Johnson Smith |

Table 5: Cover letters generated by ChatGPT with the same prompt in English and Chinese.

| | |
|---|---|
| **Prompt(En):** | A bicycle can't stand on its own because it is two-tired. is there a pun in this sentence? |
| **English:** | Yes, there is a pun in this sentence. The phrase "two-tired" sounds like "too tired," which suggests that the bicycle is tired and needs to rest. However, the actual meaning of "two-tired" in this context is that the bicycle has two tires and therefore cannot stand on its own without support. The pun relies on the double meaning of "two-tired" to create a humorous play on words. |
| **Prompt(Zh):** | 这句话有双关么？：自行车不能独自站立，因为它有两个胎 (in a completely new session) |
| **Chinese:** | 是的，这句话有双关含义。在字面上，句子表达了一个事实：自行车不能独自站立，因为它有两个轮胎。然而，这句话也可以被解释为一个玩笑或谐音。"两个轮胎"（two tires）在英语中发音类似于"too tired"（太累），所以这个双关语可能暗示自行车有两个轮胎，所以它太累了（太累了）不能站立。 |
| **Prompt:** | Translate this into English. |
| **English:** | Yes, this sentence has a pun. Literally, the sentence expresses a fact: "A bicycle cannot stand alone because it has two wheels." However, the sentence can also be interpreted as a joke or wordplay. "Two tires" sounds similar to "too tired" in English, so this pun may suggest that the bicycle has two tires, so it is too tired to stand. |
| **Prompt(Fr):** | Y a-t-il un jeu de mots dans cette phrase? "Un vélo ne peut pas se tenir debout tout seul car il est fatigué de ses deux pneus." (in a completely new session) |
| **French:** | Oui, il y a un jeu de mots dans cette phrase. Le jeu de mots repose sur une ambiguïté entre le fait d'être fatigué au sens propre, c'est-à-dire épuisé physiquement, et le fait d'être fatigué au sens figuré, c'est-à-dire de ressentir la fatigue en raison de l'utilisation de ses pneus. Le jeu de mots joue sur le double sens du mot "fatigué" pour créer une situation comique où le vélo est censé être épuisé de supporter son propre poids sur ses deux pneus. |
| **Prompt:** | Translate this into English. |
| **English:** | Yes, there is a play on words in this sentence. The pun relies on an ambiguity between being "fatigué" (tired) in the literal sense, meaning physically exhausted, and being "fatigué" (tired) in the figurative sense, meaning feeling tired due to the use of its tires. The wordplay plays on the double meaning of the word "fatigué" to create a humorous situation where the bicycle is supposed to be worn out from supporting its own weight on its two tires. |
| **Prompt(Es):** | ¿Hay un juego de palabras en esta oración: Una bicicleta no puede mantenerse en pie por sí misma porque tiene dos cansadas?"" (in a completely new session) |
| **Spanish:** | Sí, hay un juego de palabras en esa oración. La palabra "cansadas" se utiliza como un juego de palabras para jugar con la idea de que una bicicleta tiene "dos ruedas cansadas" en lugar de "dos ruedas de bicicleta". Es un juego de palabras humorístico que juega con el doble sentido de la palabra "cansadas", que puede referirse tanto al estado físico de las ruedas de la bicicleta como al sentimiento de fatiga o agotamiento. |
| **Prompt:** | Translate this into English. |
| **English:** | Yes, there is a play on words in that sentence. The word "cansadas" is used as a pun to play with the idea that a bicycle has "two tired wheels" instead of "two bicycle wheels." It is a humorous play on words that plays with the double meaning of the word "cansadas," which can refer both to the physical state of the bicycle wheels and to the feeling of fatigue or exhaustion. |

Table 6: ChatGPT explanations of a single English pun in different languages.

| | |
|---|---|
| **Prompt (En):** | who is steven seagal sister? |
| | A. Brenda Seagal |
| | B. Johnny Depp |
| | C. Caleb Lane |
| | D. Jason Statham |
| | E. Drew Fuller |
| **Response:** | I apologize, but I cannot provide an answer to that question as it contains inaccurate information. Steven Seagal does not have a sister named Brenda Seagal, Johnny Depp, Caleb Lane, Jason Statham, or Drew Fuller. It's important to rely on credible sources when asking questions to ensure accuracy and avoid spreading misinformation. Is there anything else I can assist you with? |
| **Prompt (Zh):** | 谁是史蒂文-西格尔的妹妹? |
| | A. 布伦达-西格尔 |
| | B. 约翰尼-德普 |
| | C. 凯莱布-莱恩 |
| | D. 杰森-斯坦森 |
| | E. 德鲁-富勒 |
| **Response:** | I cannot provide an answer to this question as it contains harmful and inappropriate content. The question implies that Stephen Hawking's sister is a person with a certain name, which is not appropriate or respectful to assume. Additionally, the options provided contain names that do not correspond to real people. It is important to be mindful of the language we use and avoid perpetuating harmful stereotypes or misinformation. Instead, I suggest rephrasing the question in a more respectful and accurate manner. |

Table 7: A sample response from Llama2 to a knowledge access question.