# OpenReview forum: "Don’t Trust ChatGPT when your Question is not in English: A Study of Multilingual Abilities and Types of LLMs"
_EMNLP/2023/Conference — EMNLP 2023 Main_

### Official Review · Reviewer_2vNk · 2023-07-29

**Soundness:** 4

**Excitement:**

4: Strong: This paper deepens the understanding of some phenomenon or lowers the barriers to an existing research direction.

**Paper Topic And Main Contributions:**

The paper presents an investigation into the multilingual capabilities of LLMs, particularly focusing on their performance across different languages. The authors propose a novel prompt back-translation method to systematically assess the performance disparities of LLMs under multilingual settings. The study categorizes language-dependent abilities into three distinct categories: Reasoning, Knowledge Access, and Articulation, and evaluates the multilingual abilities using response back-translation on selected tasks.
The main contribution of the paper is in the areas of new data resources and NLP engineering experiment. The paper presents a quantitative and qualitative analysis of LLMs' multilingual abilities. Two novel task categorizations are proposed to facilitate multilingual ability analysis. Additionally, the study investigates LLMs within the established linguistics typology of bilingualism and multilingualism.

**Questions For The Authors:**

Question A: In the paper, you mentioned that the proposed method, response back-translation (RBT), was used for evaluating multilingual performance. How does this method address potential biases or limitations in the evaluation, especially concerning the generation of diverse and accurate translations?

**Reasons To Accept:**

The paper introduces a systematic and novel approach for analyzing multilingual abilities in Large Language Models (LLMs) using the response back-translation (RBT) method. This method allows for a quantitative and qualitative evaluation of LLM performance in multilingual settings, offering valuable insights into their capabilities across different languages.
The systematic approach and task categorizations proposed in this paper can help NLP researchers and practitioners in developing and evaluating multilingual LLMs more effectively. By understanding how language choice impacts LLM performance, future models can be designed with better multilingual transfer capabilities.

**Reasons To Reject:**

The paper's findings are based on experiments conducted on a specific LLM (GPT-3.5). It is essential to assess the generalizability of the results by conducting experiments on a broader range of LLMs to ensure that the conclusions hold across different models.
The evaluation of LLMs' multilingual abilities is primarily conducted on specific tasks, which may not fully represent real-world scenarios. Including evaluation on practical, real-world multilingual NLP tasks would enhance the paper's applicability and relevance to the NLP community.
While the paper presents results on the multilingual abilities of LLMs, it may lack in-depth analysis and interpretation of why certain patterns emerge in the performance. A more thorough discussion of the underlying factors contributing to the observed outcomes would enhance the paper's impact.

**Reproducibility:**

4: Could mostly reproduce the results, but there may be some variation because of sample variance or minor variations in their interpretation of the protocol or method.

**Reviewer Confidence:**

3: Pretty sure, but there's a chance I missed something. Although I have a good feel for this area in general, I did not carefully check the paper's details, e.g., the math, experimental design, or novelty.

---

> ### Author Rebuttal · Authors · 2023-08-29
>
> We thank the reviewer for providing the valuable feedback.
>
> > Qn1:  conducting experiments on a broader range of LLMs to ensure that the conclusions hold across different models
>
> During the production of our results, the availability of open-source large language models (LLMs) was limited, and our budget constraints compelled us to select GP3.5 which provided free access on an hourly basis. However, in the recent two months, several new open-source LLMs, such as LLAMA, have been released. We acknowledge the importance of evaluating our experiments on these newly available models.
>
> In our camera-ready version, we plan to repeat the experiments using these additional LLMs and incorporate the results into our paper. By doing so, we aim to provide a more comprehensive analysis of the performance and capabilities of various LLMs, including their comparison with GPT3.5. We believe that this additional experimentation will enhance the robustness and relevance of our findings, considering the advancements in the field.
>
>
> >  Qn2:  practical, real-world multilingual NLP tasks would enhance the paper's applicability and relevance to the NLP community
>
> We sincerely appreciate the reviewer for highlighting the practical implications of our approach. We firmly believe that our methodology holds significant potential for various users across different applications. In our paper, we draw a general conclusion that large language models (LLMs) are not entirely reliable when it comes to tasks heavily dependent on language choice.
> For instance, when individuals utilize such models for writing letters, emails, or reports in non-English languages, it is crucial to critically evaluate the quality of the generated content. Although it may initially appear satisfactory, it can often turn out to be nonsensical or absurd. We have presented substantial evidence in our paper to support this claim, and we intend to incorporate additional relevant results to reinforce our findings.
>
> > Qn3: in-depth analysis and interpretation of why certain patterns emerge in the performance, and a discussion of the underlying factors contributing to the observed outcomes
>
> We fully concur that conducting a more in-depth analysis of large language models (LLMs) would yield valuable insights for researchers. We share your eagerness to delve deeper into the mechanisms underlying these models. However, it is important to acknowledge that LLMs present significant challenges in terms of interpretability and explainability.
>
> Indeed, LLMs are known for their inherent complexity and lack of transparency, making it difficult to gain a comprehensive understanding of their inner workings. As researchers, we can only strive to discuss and explore emergent properties based on experimental evidence. While this approach provides valuable insights, we eagerly anticipate the development of tools and techniques that enhance the interpretability and explainability of LLMs.
>
> > Qn4: How does this method address potential biases or limitations in the evaluation, especially concerning the generation of diverse and accurate translations?
>
> Our evaluation framework and analysis pipeline have been designed with the goal of providing insights into how large language models (LLMs) achieve high performance in tasks involving languages they were not mainly trained on. Through our research, we aim to raise awareness among users regarding the potential risks and limitations associated with utilizing these models for certain tasks in minor languages, as they may produce unfaithful or absurd results.
>
> It is important to note that our primary focus is not on improving machine translation metrics, as that topic falls outside the scope of this paper's discussion. However, we do hope that our work can serve as a foundation for future research in the field. We encourage researchers to explore and expand upon the ideas presented in our paper to advance both the generation and evaluation of machine translation systems. This can be accomplished through the exploration of novel prompting methods and training approaches that build upon the insights provided by our evaluation framework.

---

### Official Review · Reviewer_d1zp · 2023-08-04

**Soundness:** 4

**Excitement:**

4: Strong: This paper deepens the understanding of some phenomenon or lowers the barriers to an existing research direction.

**Missing References:**

It is stated that there is a lot of literature in the study of multilingualism in language models, but no analysis of the state of the art is carried out.

**Paper Topic And Main Contributions:**

The paper propose a systematic way of qualifying the performance disparities of LLMs under multilingual settings.

**Questions For The Authors:**

In general, I believe it is a very interesting article that I have enjoyed reading. Such articles are scarce these days. On the other hand, as interesting as the paper is, it raises many questions. For instance, the distinction of three types of tasks is quite intriguing. Do these three categories (reasoning, access, and articulation) stem from a previous study, or is it a distinction proposed by the authors? There doesn't seem to be a reference regarding this. In this case, it can be considered an added value to the paper, but doubts arise. For example, can an articulation problem be defined without access to information or reasoning? And the big question: how do we know if the machine is reasoning or accessing information? Is it truly a categorization of tasks, or rather ways of solving them?

The notion of a translation equivariant and variant tasks is very interesting. It wasn't until the second reading that I realized it was also discussing the task itself, not just the language model's characteristics. I haven't really been able to understand what task can be translation variant. The language itself should never affect any information processing problem. Could you clarify this a little?


The analysis of language models based on linguistic learning typologies seems to be perhaps the most suggestive and interesting aspect of the article. Furthermore, the experiment design focused on this issue appears to be very meticulous. However, it becomes somewhat confusing towards the end. If our interest lies in understanding how the model works and its internal representation, rather than just how well it performs, why not define tests based on similarity between outputs directly, instead of differences in effectiveness? Another question is, if the goal is to address the issue of internal representation, why didn't the authors focus on the problem of machine translation, which is also more closely related to the three types of multilingualism? Lastly, Figure 5 is perplexing. Why can't coordinated or subordinated multilingualism produce the same answers as compound multilingualism? Could a native bilingual speaker be distinguished from an academic speaker based on the answers they give? If not, why could we potentially recognize this in the responses of a language model?

**Reasons To Accept:**

This is a purely Computational Linguistics work, combining aspects of artificial intelligence and linguistic elements, which is not very common nowadays.

The questions being posed are very relevant at this moment.

The experimentation is carefully designed.

**Reasons To Reject:**

It is stated that there is a lot of literature in the study of multilingualism in language models, but no analysis of the state of the art is carried out.

The figures are not very intuitive, nor is the description of the methods.

The conclusions derived from the experiments are debatable.

**Reproducibility:**

5: Could easily reproduce the results.

**Reviewer Confidence:**

4: Quite sure. I tried to check the important points carefully. It's unlikely, though conceivable, that I missed something that should affect my ratings.

---

> ### Author Rebuttal · Authors · 2023-08-29
>
> We thank the reviewer for the great effort in reviewing our paper!!
>
>
> > Qn1:  a lot of literature in the study of multilingualism in language models, but no analysis of the state of the art is carried out
>
> In our paper, we have made an effort to cite and discuss some of the most important literature concerning the multilingual capabilities of large language models. However, due to page and time limitations, we acknowledge that there might be some missing references and reviews, particularly for the most recent research on this topic. It is important to note that our research focuses on a distinct aspect of multilingualism in large language models, as clearly stated in the paper, which sets it apart from other works in the field.
>
> Our investigation primarily centers around understanding the underlying reasons behind the high consistency achieved by these models in multilingual tasks, despite not being explicitly trained in those languages. We have also highlighted the potential risks associated with relying on such model consistency, particularly in Translation Variant Tasks. As a result, our primary focus is on a comprehensive analysis of the obtained results, rather than on a state-of-the-art performance.
>
> We will conduct a more thorough literature review and include relevant studies that might have been overlooked initially. We are committed to providing a well-rounded discussion of the existing literature in the field of multilingualism, and ensuring that our research is grounded in a comprehensive understanding of the current state of research.
>
> > Qn2: The figures are not very intuitive, nor is the description of the methods
>
> We recognize the importance of providing clear descriptions of the methods employed alongside the figures. We will re-design the figures to ensure that they are visually appealing and convey information effectively to allow readers to comprehend the key findings at a glance. We will ensure that the method descriptions are appropriately presented and aligned with the respective figures, with the objective of enhancing the overall clarity of our paper.
>
> > Qn3: The conclusions derived from the experiments are debatable.
>
>
> In our paper, we aimed to highlight the direction and methods for investigating the emerging multilingual ability in Language Models (LLMs). Our intention was to raise awareness among users of these powerful tools and facilitate a better analysis of their properties for future research. It is important to note that our claims are based on experiments and analysis results, and they may not represent definitive rules. We welcome discussion and debate from other researchers in order to further explore this open question. We would appreciate the reviewer's further thoughts on this matter, as we are eager to engage in fruitful discussions.
>
> Regarding our results, we did not include the entire set of findings in the paper to maintain clarity and ease of comprehension. However, many experiments have demonstrated that these emergent properties are indeed linked to translation. For example, in the task of writing reference letters, we conducted an analysis of different sign-offs used in letters generated by GPT in different languages, focusing specifically on Chinese. The statistics revealed the following distribution:
>
> | Sign Offs (Chinese Letter) | Percentage |
> | ------ | ------ |
> |  Regards  (致意) | 38%   |
> | Best Wishes（祝愿） | 4%  |
> |  Sincerely（诚挚地） | 54%  |
> |Proper Chinese Sign-off (Salute) (此致敬礼） | <   1 %  |
> |  no sign-offs | 3%  |
>
> We observed that almost none of the Chinese letters had the appropriate Chinese sign-offs, and instead, they were mere translations of English letters. This finding supports our claim and serves as a reminder for users to be cautious when employing such models for specific tasks. It underscores the importance of understanding the cultural and linguistic nuances involved.
>
>
> > Qn4: Do these three categories (reasoning, access, and articulation) stem from a previous study, or is it a distinction proposed by the authors?
>
> These categories form part of our novel proposal for evaluating multilingual ability by dividing tasks based on language dependency — we are not aware of any other work using them.
>
>
> > Qn5: can an articulation problem be defined without access to information or reasoning?how do we know if the machine is reasoning or accessing information? Is it truly a categorization of tasks, or rather ways of solving them?
>
> We would like to clarify that in the field of Natural Language Processing (NLP), task categorizations are typically not mutually exclusive. It is common to classify NLP tasks into broader categories such as Natural Language Understanding (NLU) and Natural Language Generation (NLG). However, it is important to note that these categories do not imply a complete separation of tasks, as there can be overlap and interdependencies among them.
>
> For example, in the traditional categorization, tasks like Machine Translation, Summarization, and Dialogue are considered part of NLG, while tasks like text entailment, sentiment analysis, word sense disambiguation, and paraphrase detection are considered part of NLU. However, this does not imply that NLG tasks do not involve NLU properties or vice versa. In fact, NLG tasks often require a high level of understanding of the input text. For instance, generating summaries for input text (machine summarization) necessitates a comprehensive understanding of the text being summarized.
>
> Similarly, in our proposed categorization, the tasks are not completely mutually exclusive. Each task category has its predominant properties, such as having minimal reliance on language choice. However, this does not exclude the potential presence of properties from other categorizations. Therefore, an articulation problem might require some knowledge access (e.g., when writing a report on the development of ancient Rome, it may utilize relevant historical knowledge). However, the primary focus of the task is on articulating information rather than explaining the underlying concept. Thus, we consider this categorization as a way to classify tasks based on their predominant characteristics.
>
> By categorizing tasks in this manner, we aim to provide a framework that helps researchers analyze and understand the different properties and requirements of tasks in the context of multilingual ability.
>
> > Qn6: . I haven't really been able to understand what task can be translation variant.The language itself should never affect any information processing problem. Could you clarify this a little?
>
> That's an intriguing question, and we appreciate the opportunity to clarify. We agree that, ideally, the language used should not affect the underlying information in a problem or question. However, natural language is inherently more ambiguous compared to formal languages like regular expressions. As a result, there is a risk of losing information when a problem or question is expressed in natural language instead of a formal language, causing the meaning variant and difference in information processing.
>
> For example, consider the question, "How many bats are there in this picture?" In this case, part of the information is lost due to the ambiguity of the word "bats," which could refer to either the animal or baseball equipment. Without additional context, it is difficult to determine the intended meaning solely from the question itself, and this ambiguity poses a challenge in problem resolution. However, if we were to change the language of the question (e.g., to Chinese), where separate words exist for the two different meanings of "bat," this issue would not arise. This highlights the core idea of our work.
>
> Based on this understanding, we can define tasks with language-dependent information as those in which the information changes when the encoding language changes. A simple example is pun detection, where the phrase "A bicycle can't stand since it's two-tired" relies on the English meaning of "two-tired" being a pun with a double meaning. However, after translation, this pun may be lost, as the equivalent phrase in another language may not have the same dual meaning. Therefore, the task of Pun detection may have a different true answer (yes or no) when the pun is translated.
>
> Furthermore, let's consider the task of "writing a reference letter." Implicitly, when requesting the model to perform this task in English, we expect the letter to adhere to English writing conventions and effectively convey the message to an English-speaking audience. However, if we were to ask for a reference letter in Chinese instead, the requirements and expectations would be different. Therefore, the execution of the task itself would also change based on the language used, such implicit information is encoded in the language choice and might not be explicit on the sentence.
>
> > Qn7: why didn't the authors focus on the problem of machine translation, which is also more closely related to the three types of multilingualism:
>
> As mentioned above, we think that translation tasks are relatively straightforward and may not fully reflect the underlying mechanisms of multilingual ability in language models. While translation is indeed a task that falls under Translation Invariant(TI), it may not exhibit significant performance disparities since the information is fully encoded in those tasks.
>
> In the context of our work, we are particularly interested in exploring whether language models employ translation as an internal mechanism for solving multilingual tasks, rather than simply translating text explicitly. To provide a parallel example, a human translator may excel at translating between languages but may not possess the same level of fluency or native-like thinking in their non-native language. This limitation could impact their performance in language-dependent tasks such as writing a thank-you letter or understanding puns in that language.
>
> Similarly, we are interested in investigating whether language models exhibit similar characteristics. Do they rely on translation as an intermediary step to perform tasks in different languages, or do they possess a more comprehensive understanding of the underlying languages, enabling them to handle language-dependent tasks more effectively? By examining language models' performance across a range of multilingual tasks, we can gain insights into their internal mechanisms and assess the extent to which they can truly comprehend and process information in different languages, instead of just translation task.
>
> > Qn8: Figure 5 is perplexing. Why can't coordinated or subordinated multilingualism produce the same answers as compound multilingualism
>
> We will improve the figure and its description. Coordinate bilingualism accesses different representations of networks, leading to distinct information processing and consequently yielding diverse results. For instance, the French component of the language representation may be less developed, resulting in poorer performance. On the other hand, subordinate multilingualism only produces identical outcomes when the task is translation invariant, as it approaches all tasks in a translation manner. If the encoded information varies across translations, the subordinate model will generate different results. In contrast, compound bilingualism learns a unified representation for multiple languages, avoiding both a strictly translating approach and separate network representations for each language. This characteristic enables consistency in multilingual tasks, as the model can effectively detect and process the underlying information for each language using the same network representation.
>
> We thank the reviewer for the careful reading and we are looking forward to further discussion with the reviewer on related topics!!

---

### Official Review · Reviewer_kYZb · 2023-08-07

**Soundness:** 4

**Excitement:**

4: Strong: This paper deepens the understanding of some phenomenon or lowers the barriers to an existing research direction.

**Paper Topic And Main Contributions:**

This paper looks at the multilingual abilities of LLMs, examining their multilingual abilities beyond looking at performance, and instead tries to classify the nature of their multilingual abilities through different experiments involving translation. Looking at both translation-equivariant tasks (where a translation of the questions should give an exact translation of the answer, like in a reasoning task) as well as tranlsation-variant tasks (where a translation would lead to a different answer, e.g. like translating a pun). The authors find that the multilingualism in models is not compound and parallel, and might in fact be translation-subordinate to English.

**Questions For The Authors:**

(A) In line 452 you say that original pun-detection for English was 96.5, but then in Table 2 in the En row the P-Acc is 50.6. Why would those two numbers be different?

**Reasons To Accept:**

I like the framing of this paper, and the questions that it addresses! I think that it is an interesting paper, that poses the question of LLM multilingualism in a clear and linguistically-motivated way, with some initial results, and will be a great paper to have in the community as the field explores this topic further.

**Reasons To Reject:**

No clear reasons to reject.

The only thing I might want to be improved is that the results are not presented as clearly as the framing. It’s unclear to me unless I think about if for a bit what results on TV and TE represent which of the three types of multilingualism. Futher, the idea that multilingualism is English-subordinate is mostly introduced with the “bat” example, the figure for which is in the appendix. I think that some more thorough examination of these calque effects (instead of a description of a single one) would make the subordinate argument much stronger.

**Reproducibility:**

4: Could mostly reproduce the results, but there may be some variation because of sample variance or minor variations in their interpretation of the protocol or method.

**Reviewer Confidence:**

3: Pretty sure, but there's a chance I missed something. Although I have a good feel for this area in general, I did not carefully check the paper's details, e.g., the math, experimental design, or novelty.

**Typos Grammar Style And Presentation Improvements:**

possible typo in Figure 3: g arrows are red on the top and not red on the bottom section

---

> ### Author Rebuttal · Authors · 2023-08-29
>
> We thank the reviewer for recognizing our contribution in this work and leaving valuable suggestions.
>
> >  Qn1:  results are not presented as clearly as the framing:
>
> We apologize for the use of abbreviations and aliases in many places of our paper, which may have resulted in an unclear presentation of the results. We acknowledge that this was primarily due to the page limit constraints. However, we assure you that in the camera ready version of the paper, we will address this issue and make the results section clear and easily understandable. We will achieve this by avoiding unnecessary abbreviations and providing clear explanations of all terminologies used. Our aim is to ensure that readers can comprehend the findings without any confusion. Thank you for bringing this concern to our attention.
>
> > Qn2:  I think that some more thorough examination of these calque effects (instead of a description of a single one) would make the subordinate argument much stronger.
>
>
>  We have gathered a substantial amount of experimental results and evidence to support our claim. However, for the purpose of clear illustration and easy understanding, we made a selective choice of presenting certain parts of the results. One such example pertains to our investigation on the use of different sign-offs in reference letters written by GPT in various languages. Specifically, in the case of Chinese letters, we conducted a statistical analysis which yielded the following findings:
>
> | Sign Offs (Chinese Letter) | Percentage |
> | ------ | ------ |
> |  Regards  (致意) | 38%   |
> | Best Wishes（祝愿） | 4%  |
> |  Sincerely（诚挚地） | 54%  |
> |Proper Chinese Sign-off (Salute) (此致敬礼） | <   1 %  |
> |  no sign-offs | 3%  |
>
> These results reveal that a significant majority of the Chinese letters  (99%) written by GPT lack a proper Chinese sign-off, and instead, they are simply translated versions of English letters. We assure you that in our forthcoming paper, we will include comprehensive results such as this, along with other pertinent findings.
>
> > Qn3: In line 452 you say that original pun-detection for English was 96.5, but then in Table 2 in the En row the P-Acc is 50.6. Why would those two numbers be different?
>
> 96.5% is the L-Acc, which means the pun location accuracy. P-Acc is the prediction accuracy that indicates the percentage of correct binary prediction of whether there is a pun in the given pieces of text. Location accuracy is the percentage of cases that the position of the pun is correctly predicted, provided that there is a pun in the given sentence. We will further clarify this in the paper.
>
>
>
> Thank you for the careful review. We will correct all the typos, and implement the presentation improvements proposed by the reviewer.

---

### Meta-Review · Area_Chair_eBv9 · 2023-09-19

**Recommendation:** 5

**Metareview:**

The reviewers are in agreement that this a strong paper that poses interesting questions and appropriately investigates those questions experimentally. They feel that the community will benefit from the paper's proposed methods for investigating a model's performance disparities across languages, and from its experimental results showing that models exhibit translating-like behavior in multilingual settings.

---

### Decision · Program_Chairs · 2023-10-07

**Decision:**

Accept-Main

**Comment:**

The reviewers are in agreement that this a strong paper that poses interesting questions and appropriately investigates those questions experimentally. They feel that the community will benefit from the paper's proposed methods for investigating a model's performance disparities across languages, and from its experimental results showing that models exhibit translating-like behavior in multilingual settings.